# Composting Sugarcane Filter Mud with Different Sources Differently Benefits Sweet Maize

Muhammad Salman [1], Inamullah [1], Aftab Jamal [2,*], Adil Mihoub [3], Muhammad Farhan Saeed [4], Emanuele Radicetti [5], Iftikhar Ahmad [4], Asif Naeem [6], Jawad Ullah [1] and Silvia Pampana [7,*]

1   Department of Agronomy, Faculty of Crop Production Sciences, The University of Agriculture, Peshawar 25130, Pakistan
2   Department of Soil and Environmental Sciences, Faculty of Crop Production Sciences, The University of Agriculture, Peshawar 25130, Pakistan
3   Center for Scientific and Technical Research on Arid Regions, Biophysical Environment Station, Touggourt 30240, Algeria
4   Department of Environmental Sciences, COMSATS University Islamabad, Vehari Campus, Vehari 61100, Pakistan
5   Department of Chemical, Pharmaceutical and Agricultural Sciences (DOCPAS), University of Ferrara, 44121 Ferrara, Italy
6   Institute of Plant Nutrition and Soil Science, Kiel University, Hermann-Rodewald-Strasse 2, 24118 Kiel, Germany
7   Department of Agriculture, Food and Environment (DAFE), University of Pisa, 56124 Pisa, Italy
*   Correspondence: aftabses98@gmail.com (A.J.); silvia.pampana@unipi.it (S.P.)

**Abstract:** Reusing organic wastes in land applications would enhance the recovery of resources, following the concepts of the circular economy. The sugarcane-based sugar industry produces various by-products (e.g., sugarcane filter mud, molasses, and bagasse) that have the potential to contribute to crop production and soil fertility, owing to their high contents of organic matter and nutrients. Although the agricultural benefits of compost utilization in agriculture have been well-documented, to the best of our knowledge, few scientific data are currently available on the effects of sugarcane filter mud combined with the application of compost for increasing crop production. Thus, a field experiment was carried out to study how sugarcane filter mud, in combination with two compost sources, affected the growth and yield of sweet maize (*Zea mays* var. *saccharata*). We compared (i) two types of compost made from brassica residue and household waste applied at a rate of 9 t ha$^{-1}$, and (ii) two application rates of sugarcane filter mud: 0 and 2 t ha$^{-1}$ to two controls without any compost application: one with (+SFM) and one without (−SFM) sugar filter mud. The results highlighted that all crop growth and yield parameters benefited more from the domestic waste compost than from the brassica straw compost. Moreover, the addition of sugar filter mud to the compost further boosted the crop performance. Based on the above results, we concluded that the addition of sugarcane filter mud to locally available composts is a feasible approach for more sustainable production of sweet maize, combining efficient waste disposal and the provision of organic matter to the soil.

**Keywords:** circular economy; filter mud; composts; household waste; crop productivity; managing bio-waste; organic farming; sugar industry wastes; sustainability; yield; waste disposal; *Zea mays* L.

## 1. Introduction

Sugarcane (*Saccharum officinarum* L.) is a perennial crop belonging to the *Poaceae* family that is grown in tropical and subtropical regions around the world, mainly for sugar production. Brazil, India, Thailand, China, and Pakistan are the major producers, accounting for more than 70% of world production [1]. In Pakistan, sugarcane is the second largest cash crop and cultivated on about 1 million hectares, with a yield of 55 tons per hectare, and the sugar industry contributes about 4% to the country's gross domestic product [2]. Sugar is produced by extraction from the juice obtained from crushed canes

after cleaning and cutting. During sugarcane processing, large amounts of various by-products, namely sugarcane bagasse, bagasse ash, press mud cake, vinasse, and spent are released. After the extraction of the sugarcane juice, bagasse is the remainder, while sugarcane filter mud (SFM) is the cake residue formed from the precipitated impurities after removal by filtration. Accordingly, filter mud production can vary from 1 to 7% of the total weight of sugar cane [3]. As sugarcane production in Pakistan in 2022/23 is forecast to be 89.5 million metric tons [4], with an average yield of 3%, the country's output of fresh filter press mud is estimated to be about 2.7 million t. The disposal of such a large amount of this by-product is a major environmental and economic issue. Burning is a common practice, but the nutrients contained are released into the environment. Conversely, reusing SFM through land applications would enhance the recovery of resources, according to the concepts of the circular economy [5], because it would provide important plant nutrients to the soil and have beneficial effects on soil texture, water holding capability, soil porosity, hydraulic properties, and soil bulk density [6–8]. In heavy soils it improved aeration and drainage, while in sandy soils, it was beneficial for moisture conservation [9]. Filter mud generally has an average moisture content of 50–70%, ideal for soil microorganisms and earthworms [10], and is rich in organic carbon, nitrogen (N), phosphorus, potassium, and micronutrients [11]. The addition of filter mud has been proven to increase cation exchangeability and nutrient availability. However, its composition varies with the locality, cane variety, milling efficiency, and method of clarification.

After rice and wheat, corn (*Zea mays*) is the third most produced crop in Pakistan. It accounts for 8.5% of the nation's total cereal cultivation area, 2.2% of the value added in the agriculture sector, and 0.4% of the gross domestic product (GDP) [12,13]. *Zea mays* L. var. saccharata, often known as sweet maize, is one of the numerous varieties of maize, along with flint corn, dent corn, popcorn, flour corn, and pod corn. It is grown for local markets in many parts of Khyber Pakhtunkhwa (Pakistan), including Swat, Mansehra, Swabi, and Parachinar [14]. Worldwide, the food industry uses sweet maize as a raw or processed material, and it is a plant that is grown for human consumption. It is well-liked by consumers because of its unique taste, pleasant flavor, and sweetness. Due to its nutritional qualities that promote health, sweet maize has a significant role in the human diet [15]. Sweet maize is the same botanical species as common corn, but it displays a genetic variation in the sugar conversion to starch in the endosperm, such that it develops higher sugar levels in the seeds [16]. It also has a shorter growing season and higher harvest index, because it is usually harvested earlier (i.e., commercial maturity corresponds to one-half to three fourths of grains at full maturity) than conventional corn. On the other hand, more N is removed in the harvesting of marketable ears and hence larger amounts of nutrients must be supplied for optimal production and quality. Thus, sweet maize production is facing sustainability problems, because of the indiscriminate use of chemical fertilizers and pesticides. Alternative sweet maize production methods that are more economical and sustainable, while also improving sweet maize productivity and reducing the excess use of chemical fertilizers, are therefore needed. One sustainable practice is combined soil fertility management that integrates mineral fertilization with organic waste materials available near the cultivation site, because the addition of organic matter has been proven to sustain sweet maize production [17], by improving soil properties, e.g., the structure and aggregate stability, as well as the moisture retention capacity [18]. This is particularly critical for Pakistan's soils, which are typically poor in organic matter. Here, two types of local organic wastes are most frequently available: agricultural (crop residues), which are primarily post-harvest wastes; and domestic, (household waste or residential trash) which are the waste materials produced by households. The composting of both these organic materials can reduce their weight and volume, and they can be transformed into organic fertilizers. However, the organically bound nutrients in both compost types are not immediately available for crops because they must be mineralized before they can be assimilated by plants. Conversely, the organic matter of filter mud is more soluble and quickly becomes subject to microbial activity [19,20]. Mud, as a waste residue from



sugar processing, is commonly utilized through direct land application, but we speculated that an indirect (e.g., after composting) application could better recover its fertilizer value. Hence, it was hypothesized that the application of sugarcane filter mud combined with composted organic sources could more closely meet crop requirements and differently affect the phenological characteristics, thereby improving growth and, finally, the yield and yield components. Sweet maize was selected for this research because it has relatively high N requirements and is a widespread cash crop in the region. To support the disposal of locally available wastes and integrate mineral fertilization, we evaluated two different materials (e.g., crop residues and domestic wastes), because their typology can determine the quality of the compost produced.

## 2. Materials and Methods

### 2.1. Experimental Site

A field experiment was conducted at the Agronomy Research Farm of the University of Agriculture Peshawar in Pakistan (34.01° N, 71.35° E, 350 m a.s.l.) during the spring of 2021. The climate of the area is semiarid; the mean annual precipitation ranges from 250 to 500 mm, with 60–70% occurring in summer, while the remainder falls in winter. The minimum and maximum average temperatures of the longest period are 26 °C and 37 °C, respectively [21]. The region has intensive agricultural activities with cultivation of a variety of field crops (e.g., wheat, rice, maize, tobacco, and sugarcane). The experimental site had previously been cultivated with a sweet maize–wheat–sweet maize–sugarcane rotation. Soil samples from the upper horizon (0–20 cm) were collected in each plot before the start of the experiment using a soil auger. The samples were air dried, crushed to pass through a 2 mm sieve, mixed to make a composite sample, labeled, and stored in a plastic bag. Standard laboratory methods were used for physical and chemical characterization [22]. The soil type was Calcaric Luvisols (FL ca), according to the World Reference Base (WRB) system of soil taxonomy; had a clay loam texture (sand 18%, silt 49%, and clay 33%); was alkaline (pH 8.1), calcareous (166.3 g kg$^{-1}$ lime), and non-saline (0.3 dS m$^{-1}$ electrical conductivity); with low organic matter (8.9 g kg$^{-1}$) and nutrient contents (total N 0.1 g kg$^{-1}$; AB-DTPA assimilable phosphorous and potassium 4.7 mg kg$^{-1}$ 129 mg kg$^{-1}$, respectively).

### 2.2. Experimental Design and Crop Managements

The research was carried out with a factorial design, with three replications with the following treatments: (i) two sources for compost (CS) (crop residues (brassica residue) (CS$_1$) and domestic wastes (household waste) (CS$_2$)) applied at the rate of 9 t ha$^{-1}$; (ii) two sugarcane filter mud (SFM) applications (application of SFM at the rate of 2 t ha$^{-1}$ (+SFM), and SFM not applied (−SFM)). To compare these novel SFM utilizations to the common practice of SFM application to soil without composting, we introduced two controls (CS$_0$) without any compost application: one with (+SFM) and one without (−SFM) sugar filter mud.

All treatments were applied before crop sowing. The plot size was 5 m × 4 m. Plant to plant distance was 20 cm, and the row to row distance was 70 cm, with six rows per plots. Except for the experimental treatments, all agronomic practices were performed according to the local farmers' practices and were the same in all experimental plots.

The experimental field was ploughed twice with cultivator followed by a rotavator, on 23 April 2021, and basal fertilization with 90 kg P$_2$O$_5$ ha$^{-1}$ single super phosphate was performed based on a soil analysis. Following organic source application (as per the experimental design) and tillage, sweet maize (Azam variety) was sown on 24 April 2021 at the rate of 30 kg ha$^{-1}$. N fertilizer was applied as Urea at the rate of 120 kg ha$^{-1}$, to avoid early-season N deficiency. This rate was expected to provide half of the N crop requirement, while the remainder was supplied by the organic fertilization. Weeds were controlled by hand hoeing, and the crop was irrigated at two week intervals, until the early milk stage (BBCH stage 73) [23]. Irrigation was carried out with 2540 m$^3$ ha$^{-1}$ of water from the Malakandher River. On average, the temperature of the water used was about

23 °C; the pH 6.3, EC 0.83 dSm$^{-1}$, and Cr remained below the acceptable limits of the National Environmental Quality Standards of Pakistan.

### 2.3. Compost Material and Treatment Preparation

The two organic materials (e.g., of domestic and agricultural origin), without and with SFM, were composted according to local practices. The domestic waste type was household waste (i.e., kitchen waste such as bits of food left over from cooking, such as vegetable peelings and scraps from people's plates), and the agricultural waste type was from crop residues (i.e., plant parts such as leaves and stems left after brassica were harvested). Their different starting properties lead to different composting methods, to achieve optimal control of the main factors of the composting process (e.g., moisture content, temperature, and oxygen); the windrow method was used for composting the domestic waste, while the drum method was adopted for the brassica straw. Briefly, the first composting method consisted of placing the domestic waste mixed with farmyard manure in piles (i.e., windrows) 1.5 m high and about 1.8 m wide, which were regularly turned. The turning operations rearranged the composting material and enhanced passive aeration. The second composting method involved mixing the brassica straw pieces with farmyard manure, which was then put into drums, covered with caps, and left in an open space to allow natural aeration. For compost preparation, the wastes were collected and covered with plastic sheets; at one-week intervals, water was sprinkled, and then they were again covered with plastic sheets. After two months, the compost was ready to use. The physiochemical properties and nutrient constituents of compost were as follows: brown to black color, pH = 7.5, EC = 0.7 dS m$^{-1}$, total N = 14.6 g kg$^{-1}$, total P = 5.5 g kg$^{-1}$, and total K = 9.4 g kg$^{-1}$. The moisture level was checked regularly (15 days), and the material was sprinkled with water weekly.

To establish the treatments without sugarcane filter mud application (−SFM), only raw materials were composted; conversely, for the treatments with filter mud application (+SFM treatment), filter mud was added and mixed with each source (brassica straw or domestic waste) and composted together. The sugarcane filter mud, analyzed according to [24], had the following properties: 29% humidity, pH 5.1, organic C 39.3%, organic matter 83%, and total N 1.9%.

The final physical and chemical properties of the prepared composts applied to the crops, both with and without addition of SFM, are illustrated in Table 1. Analyses were made according to international methods for the examination of composting and composts [25]).

**Table 1.** Main physical and chemical properties of the compost materials (brassica straw and household waste) with and without the addition of sugarcane filter mud.

| Properties | CS$_1$ − SFM | CS$_2$ − SFM | CS$_1$ + SFM | CS$_2$ + SFM |
|---|---|---|---|---|
| Bulk density (g cm$^{-3}$) | 0.45 | 0.56 | 0.84 | 1.89 |
| Moisture (%) | 15.6 | 20.6 | 25.5 | 40.4 |
| Porosity (%) | 50.6 | 60.5 | 72.3 | 80.3 |
| pH | 6.1 | 7.1 | 8.1 | 9.2 |
| Total carbon (%) | 50.4 | 25.7 | 70.0 | 28.9 |
| Total organic matter (%) | 27.5 | 30.6 | 39.9 | 41.3 |
| Total nitrogen (%) | 1.3 | 1.2 | 7.0 | 2.3 |

CS$_1$: compost from brassica straw; CS$_2$: compost from domestic waste; SFM: without sugarcane filter mud; +SFM: with sugarcane filter mud. Data are means of three replicates.

### 2.4. Sampling Procedures and Measurements

Weather data were recorded from an automatic station located at the experimental site at the Agronomy Research Farm of the University of Agriculture Peshawar (Pakistan) (34.01° N, 71.35° E, 350 m a.s.l.). Throughout the experiment, the main phenological phases of the sweet maize were recorded using the BBCH scale [23]. Days to emergence, silking,

tasseling, and physiological maturity were calculated by counting the days from the date of sowing to the date when emergence, silks, tassel production, and complete loss of glumes green color were observed on 80% of plants in each plot. Emergence per unit area, number of leaves per plant, plant height, and leaf area per plant were determined at the milk stage, BBCH 75 (11 June 2021). Sweet maize yield and related trails were also recorded at final harvest, BBCH 89 (24 July 2021). Ear weight was calculated by weighing the cob without the husk on a weight balance and measuring the ear length, and five plants from each plot were randomly selected and then averaged. To count the number of grains per ear, five cobs were randomly selected, the grains were counted separately, and their average was calculated. The number of seedlings in each plot was calculated with a meter rod at three different places and the emergence m$^{-2}$ was calculated using the formula below.

$$\text{Emergence } \left(\text{n m}^{-2}\right) = \frac{\text{Total number of seedlings emerged (n)}}{\text{Row to row space (m)} \times \text{Number of rows (n)} \times \text{row length (m)}} \quad (1)$$

For the number of leaves per plant and plant height, five plants were chosen at random, and the number of leaves per plant was counted and then averaged on each plot. Plant leaf area was calculated by multiplying the leaf length and width by a correction factor (CF, 0.65), using the formula in [26].

$$\text{Leaf area } \left(\text{cm}^2\right) = \text{Average of leaf area } \times \text{Number of leaves per plant } \times \text{CF} \quad (2)$$

Thousand grain weight (g) was recorded using an electronic balance and counting a thousand grains from each plot at random. The biological yield was measured by harvesting four central rows in each plot and calculated as in Equation (3) [27]. The harvested material was sun dried and weighed. Grain yield and harvest index were calculated using Equation (4) and Equation (5), respectively [27], while shelling (%) used Equation (6).

$$\text{Biological yield } \left(\text{t ha}^{-1}\right) = \frac{\text{Total plant weight in 4 central rows}}{\text{R} - \text{R distance (m)} \times \text{Row length(m)} \times \text{No. of rows}} \times 10 \text{ m}^2 \quad (3)$$

$$\text{Grain yield } \left(\text{t ha}^{-1}\right) = \frac{\text{Biological yield in 4 central rows}}{\text{R} - \text{R distance (m)} \times \text{Row length(m)} \times \text{No. of rows}} \times 10 \text{ m}^2 \quad (4)$$

$$\text{Harvest index (\%)} = \frac{\text{Grain yield}}{\text{Biological yield}} \times 100 \quad (5)$$

$$\text{Shelling percentage (\%)} = \frac{\text{Grain weight (g)}}{\text{Cob weight (g)}} \times 100 \quad (6)$$

*2.5. Statistical Analysis*

The collected data were statistically analyzed with a factorial design using analysis of variance (ANOVA) in the statistical package Statistix8.1 (Statistix8.1, Tallahassee, FL, USA). If the F-values were significant, means were then compared with the LSD test at an $\alpha = 0.05$ level of probability [28]. Furthermore, a canonical discriminant analysis (CDA) was used to identify relationships between each studied trial and the fertilization management practices of the sweet maize. The CDA analysis results are presented in a two-dimensional canonical discriminant structure plot.

**3. Results**

*3.1. Weather Conditions*

Throughout the sweet maize growing season (from 24 April to 24 July), air temperatures followed the long-term average and increased from 30 °C in April to 40 °C in June, and then slightly decreased to 35 °C in July (Figure 1). Total rainfall was 190.4 mm, with most falling in July (122.2 mm, Figure 1).

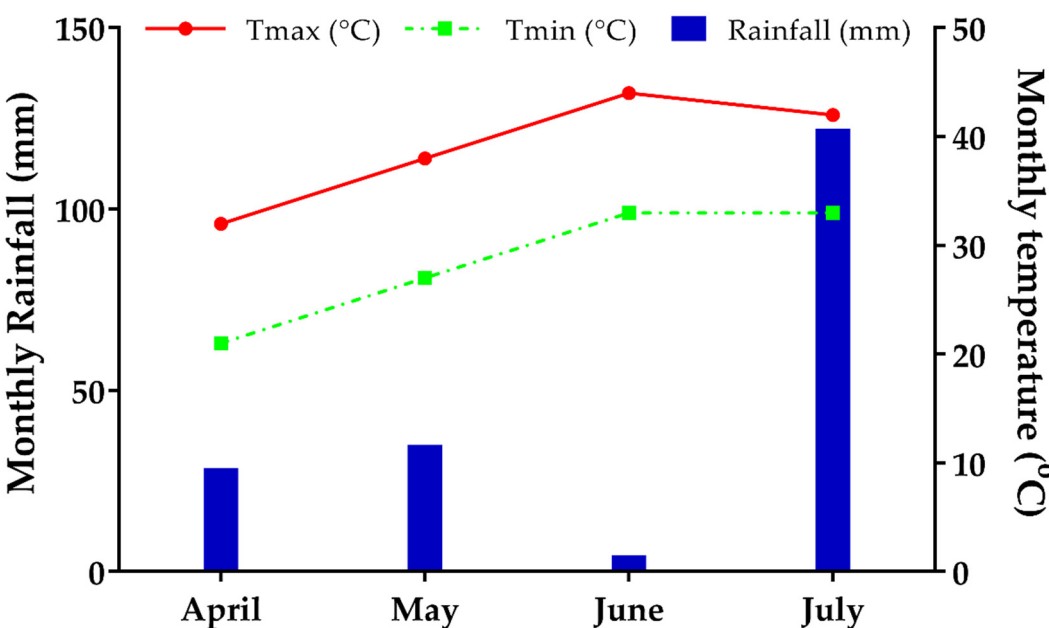

**Figure 1.** Monthly rainfall, and minimum and maximum air temperatures during the sweet maize growing season (April–July 2021) in Peshawar (Pakistan).

### 3.2. Phenological and Growth Parameters

The data evidenced a significant effect of the compost source on the days to emergence of sweet maize ($p = 0.0018$, Figure 2A), while the sugarcane filter mud and the interactions between sugarcane filter mud and compost source (SFM × CS) was not significant. Increased days to emergence (7.5 days) were recorded in the plots where domestic waste ($CS_2$) was applied compared to the brassica residue compost ($CS_1$). Among the compared fertilization strategies (SFM × CS), delayed emergence (7.7 days) was reported when SFM (at the rate of 2 kg ha$^{-1}$) was applied in combination with brassica residue compost ($CS_1$) (at the rate of 9 t ha$^{-1}$); whereas an earlier emergence (5.3 days) was recorded in the plots that received only SFM (Figure 2A). A significant effect of the compost source (at $p < 0.001$) and its interaction with SFM application SFM × CS ($p < 0.05$) was noted in days to tasseling, while SFM was found to be non-significant (Figure 2B). The mean data showed that the maximum days to tasseling (56.3 days) were recorded in plots where domestic waste compost ($CS_2$) was applied, while brassica waste compost ($CS_1$) recorded the minimum days to tasseling (54.8 days). Increased days to tasseling (56.7 days) were reported when sugarcane filter mud was applied in combination with domestic waste compost ($CS_2$), while minimum days to tasseling (51.7 days) were recorded with the only SFM application.

ANOVA revealed that the application of different compost sources significantly affected the days to silking of sweet maize ($p < 0.001$), while SFM application and the interactions of SFM × CS were non-significant (Figure 2C). The mean data showed that the maximum days to silking (60.7 days) were observed with domestic waste compost ($CS_2$), while brassica residue compost ($CS_1$) showed less days to silking (59.6 days). The days taken by plants to reach physiological maturity was influenced by both the compost source ($p < 0.001$) and interaction of SFM × CS ($p < 0.05$), while the application of SFM was non-significant (Figure 2D). Increased days to physiological maturity (96.8 days) were recorded in plots where domestic waste compost ($CS_2$) was applied compared to plots where no compost was applied ($CS_0$) (90.5 days). The maximum days to physiological maturity (97.7 days) were recorded with co-application of domestic waste compost with sugarcane filter mud ($CS_2$ + SFM), while the minimum days to physiological maturity (89.3 days) were recorded in the control plots with the only SFM application.

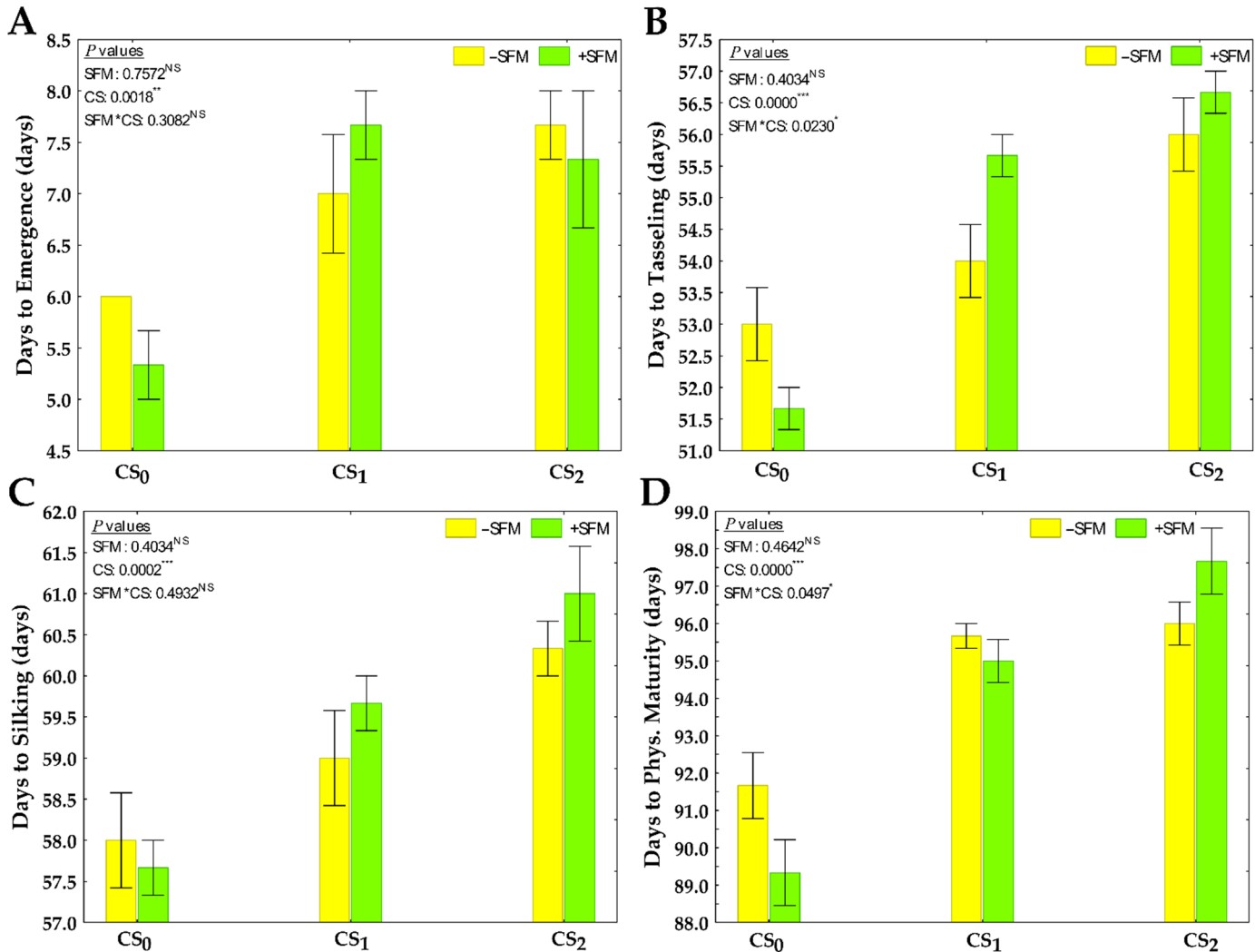

**Figure 2.** Length (days) of the main growth stages of sweet maize as affected by compost source, and sugarcane filter mud application. Days to emergence (**A**), days to tasseling (**B**), days to silking (**C**), and days to physiological maturity (**D**). $CS_0$ = control treatment with no CS input; $CS_1$ = crop residues (brassica residue); $CS_2$ = domestic wastes (household waste); $-$SFM = sugarcane filter mud not applied; $+$SFM = sugarcane filter mud applied. Bars represent $\pm$SE ($n$ = 3). * $0.01 < p \leq 0.05$, ** $0.001 < p \leq 0.01$, *** $p \leq 0.001$; NS: not significant.

Both compost sources significantly ($p < 0.001$) influenced the seedling emergence of sweet maize, but the application of SFM and the interaction of SFM $\times$ CS were found to be non-significant for this parameter (Figure 3A). More seedlings (13.0 m$^{-2}$) were recorded in plots with the application of domestic waste compost ($CS_2$), while less plants (6.0 m$^{-2}$) were observed in plots without compost application ($CS_0$). The maximum seedlings (13.0 m$^{-2}$) emerged with the combination of domestic waste compost and sugarcane filter mud ($CS_2$ + SFM), while the application of only SFM resulted in the minimum (6.0 m$^{-2}$) emergence. Analysis of the data revealed that the plant height of sweet maize was significantly influenced by the compost source ($p < 0.001$), while the application of SFM and the interaction of SFM $\times$ CS were non-significant (Figure 3B). Taller plants (192 cm) were recorded in the plots with domestic waste compost ($CS_2$), while the control plots ($CS_0$) had the shortest plants (187 cm). The combination of domestic waste compost and sugarcane filter mud ($CS_2$) +SFM) resulted in the maximum plant height of (193 cm), while the minimum plant height of (190 cm) was recorded with the sole application of SFM. Both compost sources significantly ($p < 0.001$) influenced the number of leaves plant$^{-1}$ of sweet

maize, while the application of SFM and the interaction of SFM $\times$ CS were found to be non-significant (Figure 3C). Plants grown with the application of domestic waste compost ($CS_2$) had two more leaves than the control ($CS_0$). Application of domestic waste compost and sugarcane filter mud ($CS_2$ + SFM) promoted a greater number of leaves plant$^{-1}$ (10) than the sole application of SFM (8). Similarly, the leaf area of sweet maize was significantly ($p < 0.001$) higher with domestic waste compost ($CS_2$) (22.3 %) compared to the control ($CS_0$). Application of SFM and the interaction of SFM $\times$ CS did not significantly affect the crop leaf area (Figure 3D). The leaf area was increased by 19% with the co-application of domestic waste compost and sugarcane filter mud at the rate of 2 t ha$^{-1}$ ($CS_2$ + SFM), compared to only SFM.

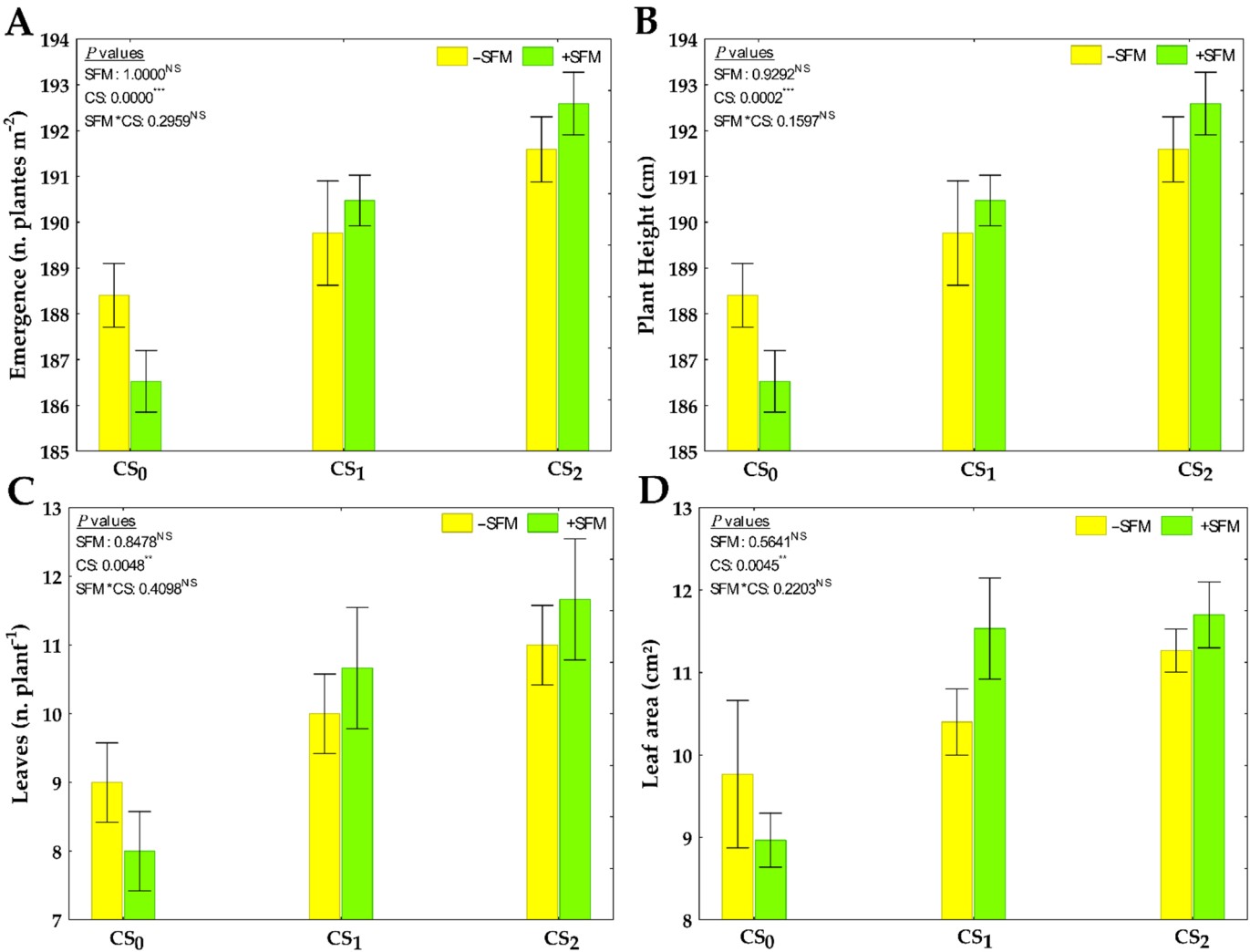

**Figure 3.** Plant emergence (**A**), plant height (**B**), number of leaves (**C**), and leaf area (**D**) of sweet maize, as affected by the compost source and sugarcane filter mud application. $CS_0$ = control treatment with no CS input; $CS_1$ = crop residues (brassica residue); $CS_2$ = domestic wastes (household waste); $-$SFM = sugarcane filter mud not applied; +SFM = sugarcane filter mud applied. Bars represent $\pm$SE ($n$ = 3). ** $0.001 < p \leq 0.01$, *** $p \leq 0.001$; NS: not significant.

### 3.3. Sweet Maize Yield and Yield Components

The ear length of sweet maize was increased by 29% with the application of domestic waste compost ($CS_2$), when compared to the control ($CS_0$). Application of SFM and the interaction SFM $\times$ CS were non-significant (Figure 4A). A total increase of 40% in ear length was recorded in the plots where domestic waste compost ($CS_2$) and sugarcane filter mud were concurrently applied at the rate of 2 t ha$^{-1}$ (+SFM), as compared to SFM

alone. A significant effect of the compost source ($p < 0.001$) and of SFM $\times$ CS (interaction $p < 0.05$) was recorded on ear weight, while SFM was found to be non-significant for the ear weight of sweet maize (Figure 4B). According to the mean data, the application of domestic waste compost ($CS_2$) increased the ear weight by 12% over the control ($CS_0$) and by 1% over the brassica waste compost ($CS_1$). Blending domestic waste compost ($CS_2$) and sugarcane filter mud (+SFM) increased the ear weight of sweet maize by 40% over sole application of SFM. The compost source significantly influenced ($p < 0.001$) the number of grains ear$^{-1}$, while SFM and the interaction of SFM $\times$ CS were not significant (Figure 4C). An increase (+8%) over the control ($CS_0$) in the number of grains ear$^{-1}$ was noted with the application of domestic waste compost ($CS_2$). Co-application of domestic waste compost and sugarcane filter mud ($CS_2$ + SFM) significantly increased the number of grains ear$^{-1}$ (11%) compared to the sole application of SFM. The application of compost also significantly ($p < 0.001$) increased the thousand grain weight of the sweet maize, while SFM and SFM $\times$ CS did not significantly change this parameter (Figure 4D). A thousand grain weight (15%) higher than the control ($CS_0$) was prompted by the application of domestic waste compost ($CS_2$). Application of both domestic waste compost and sugarcane filter mud ($CS_2$ + SFM) resulted in a higher thousand grain weight (24%) than solely SFM application. Similarly, a higher (86%) shelling percentage was recorded with domestic waste compost ($CS_2$) application than the control (79%). Again, SFM and the interaction of SFM $\times$ CS were non-significant (Figure 4E). The combination of domestic waste compost and sugarcane filter mud ($CS_2$ + SFM) showed a higher shelling percentage (87%) than the sole application of SFM.

The data showed a significant increase in biological yield, as influenced by the compost source ($p < 0.001$) and interaction of SFM $\times$ CS ($p < 0.001$), while SFM application did not significantly influence this (Figure 4F). Domestic waste compost ($CS_2$) produced higher yields (9004.2 kg ha$^{-1}$) than brassica residue compost ($CS_1$) (8317.2 kg ha$^{-1}$), while the control plants had the lowest yields (7045.7 kg ha$^{-1}$) (Figure 4F). Co-application of domestic waste compost with sugarcane filter mud produced up to 9555 kg ha$^{-1}$, while the minimum value (6631.7 kg ha$^{-1}$) was observed in the control plot with SFM application. Similarly, the variation in the grain yield of the sweet maize crop was in response to the compost ($p < 0.001$) and SFM $\times$ CS interaction ($p < 0.05$), while SFM application had a non-significant effect (Figure 4G). The domestic waste compost ($CS_2$) performed best in terms of the grain yield (2824.8 kg ha$^{-1}$), while a lower grain yield (2472.5 kg ha$^{-1}$) was recorded when brassica residue compost ($CS_1$) was used, followed by the control plots ($CS_0$) (1855.7 kg ha$^{-1}$). The highest grain yield (3047 kg ha$^{-1}$) was harvested with the co-application of domestic waste compost with sugarcane filter mud ($CS_2$ + SFM). This was partly due to a higher harvest index (32%) being recorded with the application of domestic waste compost ($CS_2$) than with the control ($CS_0$) (27%) (Figure 4H). SFM application and the interaction SFM $\times$ CS were non-significant for the harvest index of sweet maize (Figure 4H).

To obtain a multivariable view of all the parameters of the sweet maize fertilization management practices, as well as to better evaluate the mutual relations between different traits measured and to classify the contributions of these traits, canonical discriminant analysis (CDA) was performed for all yield characteristics of the sweet maize under the different fertilization management practices. Biplots of the sweet maize yield characteristics based on CDA analysis of the sugarcane filter mud $\times$ compost source interaction showed that the first two canonical variables explained 67% of the total variance (Figure 5). The data obtained from CDA application allowed the determination of the best fertilization strategy for sweet maize performance. The findings revealed that the most important yield trails for sweet maize (plant height (PH), number of grains ear$^{-1}$ (NG), 1000-kernels weight (TGW), grain yield (GY), and biological yield (BY)) were associated with +SFM $-$ $CS_2$, and +SFM $-$ $CS_1$. (Figure 5).

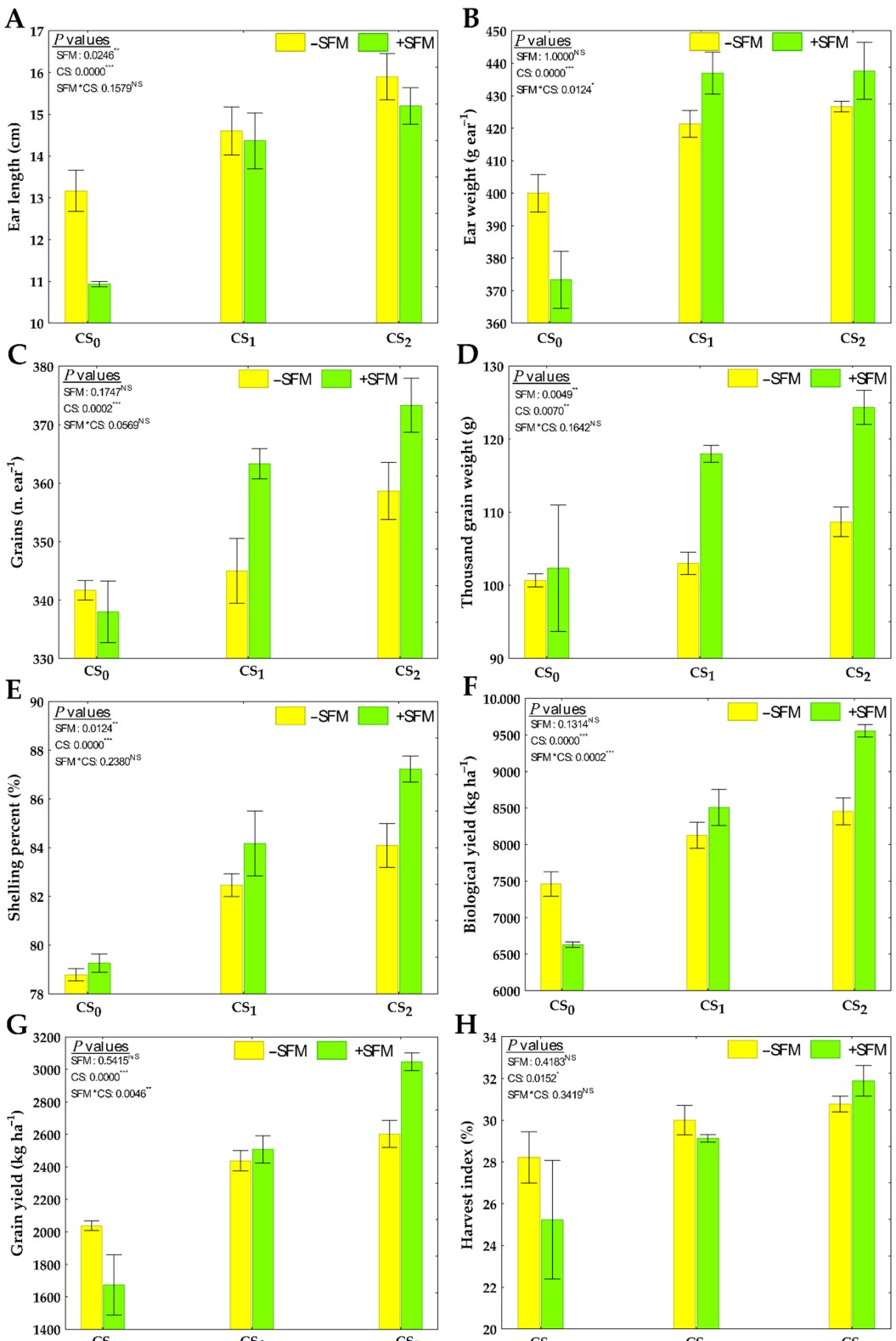

**Figure 4.** Yield components of sweet maize as affected by the compost source and sugarcane filter mud application. Ear length (**A**), ear weight (**B**), grain per ear (**C**), thousand grain weight (**D**), shelling percentage (**E**), biological yield (**F**), grain yield (**G**), harvest index (**H**). $CS_0$ = control treatment with no CS input; $CS_1$ = crop residues (Brassica residue); $CS_2$ = domestic wastes (household waste); −SFM = sugarcane filter mud not applied; +SFM = sugarcane filter mud applied. Bars represent ± SE ($n$ = 3). * $0.01 < p \leq 0.05$, ** $0.001 < p \leq 0.01$, *** $p \leq 0.001$; NS: not significant.

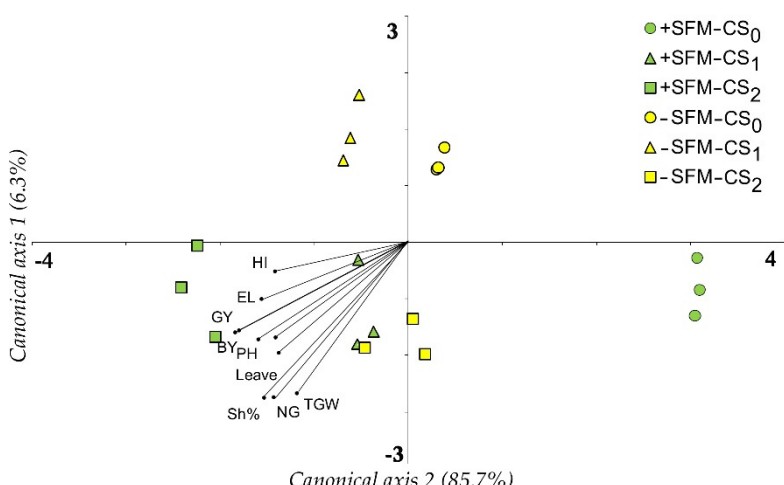

**Figure 5.** Biplot from the canonical discriminant analysis (CDA) of sugarcane filter mud and compost source effects on sweet maize yield characteristics. −SFM = sugarcane filter mud not applied; +SFM = sugarcane filter mud applied; $CS_0$ = control treatment with no CS input; $CS_1$ = crop residues (Brassica residue); $CS_2$ = domestic wastes (Household waste); PH = plant height; EL = ear length; Leave = number of leaves; LA = leaf area per plant; TGW = thousand grains weight; SH% = shelling percentage; NG = number of grain; GY = grain yield; BY = biological yield; HI = harvest index.

## 4. Discussion

Practical approaches for disposing of waste from the sugarcane industry and other locally produced wastes are urgently needed, not only to safely dispose of residues, but also to exploit them as highly valuable sources of organic matter and nutrients to sustain the circular economy [29]. Accordingly, the present research aimed to evaluate the effects of the addition of different composted organic sources and sugarcane filter mud to compost materials in local sweet maize systems in Pakistan. Our results clearly demonstrated that amending the soil with composts derived from both agricultural and domestic wastes ultimately increased crop yields, thus contributing to the long-term sustainability of these sweet maize production systems. However, sweet maize producers should carefully consider where they source these materials, because we also found that the different compost sources displayed distinct advantages, also due to the different methods of composting, as suggested by Adeduran et al. [30]. In our research, the sweet maize performed best with the application of domestic waste compost, and the effect was further improved when SFM was added. This was probably because the domestic waste ensured a higher organic matter content than the brassica straw, together with a more favorable C/N ratio and a better pH value. In addition, we found that SFM addition greatly increased the total N content of the composted materials, thus the positive response to SFM addition can likely be attributed to the higher nitrogen concentration in the final compost. Increased mineral availability was also reported by Kumar and Chopra [24]. Moreover, besides supplying plants with mineral nutrients, compost can improve soil's physical characteristics, nutrient availability, and microbial activity [31]. Thus, it is possible that other dynamics such as higher water availability and better physical and biological soil conditions could have contributed to the increase in crop growth displayed by the crops in our treatments. Among the physical characteristics, the soil organic matter content increased the soil temperature, which caused improved seedling emergence [32]. Furthermore, sweet corn development has been demonstrated to be related to temperature [33,34]. Even if domestic waste compost slightly increased the days to crop emergence, this fact was more than counterbalanced by the boosted rate of emergence, which additionally improved with SFM addition. Similarly, Raman and collaborators [35] determined that the effect of sugarcane filter mud application was stimulatory to maize germination, because of its positive effect on soil physical conditions such as soil aeration, soil water holding capacity, and bulk density. This was

even more important under the present conditions, because the temperature was higher at the experimental site and the soil had a low organic matter content, so the soil dried up quickly, and this may have determined the lower emergence of the controls.

However, if the compost had come from a superior source, such as domestic waste with a higher bulk density and porosity, the soil conditioning effect may have been furthermore enhanced. In addition, differences in the N content of the residues could also have affected the sweet corn growth, as observed for oilseed rape leaves decomposing in soil by Trinsoutrot et al. [36].

We also found that all compost materials delayed the time required for the tasseling and silking of the crops. The nitrogen content of the composts was expected to be gradually released following the organic matter mineralization, as we previously demonstrated for other organic wastes [37], and this could account for the prolonged vegetative period we observed. Time of tasseling and silking were also altered with sugarcane press mud application in the research of Muhammad et al. [38], and the present findings are further supported by those obtained by Imran and colleagues [39] with different organic fertilizations. Furthermore, organic fertilization was inferred to have improved N-resource partitioning in sweet corn by Woodruff et al. 2019 [40].

Similarly, maturity was achieved later by sweet maize plants with compost application. Thus, the boosted yields were likely due to the longer growing period, which allowed the optimal photosynthesis and nutrient uptake and led to a greater translocation of photosynthates from source to sink, due to the longer reproductive phase, as proposed by Pampana and coauthors and by Kumar et al. [41,42]. The lengthening of the period up to emergence–silking due to compost application allowed for a greater accumulation of assimilates in the maize, thus increasing the source for remobilization in the following period, together with an improved nutrient availability during the crop growth cycle [43]. Thus, our outcomes are also consistent with those of Masti et al. [44], who found that a continuous supply of nutrients and manure could cause delayed maturity.

This hypothesis was further confirmed by the increase in the number of leaves and leaf area per plant we registered, as well as the plant height, due to the use of domestic compost and SFM. Sugarcane filter mud is rich in macronutrients (nitrogen, phosphorus, magnesium, potassium, and calcium) and micronutrients (manganese, iron, and zinc) that promote plant growth and development [45,46]. Similarly, higher plants were reported by Makinde et al. with the application of by-products [47]. Moreover, Varatharajan et al. [48] described that a better leaf area index led to improved interception, absorption, and utilization of radiant energy, resulting in a faster photosynthetic rate and, ultimately, in greater dry matter accumulation by crops.

However, sugarcane filter mud addition alone ($CS_0$ + SFM) did not improve, and even reduced, sweet corn yields, because it is an insoluble material, whose decomposition in natural conditions is a slow process [49]. This finding further confirmed our hypothesis that its indirect application (i.e., composting) would make nutrients available more synchronously for crop production and better improve the soil physical characteristics. Overall, we demonstrated that the addition of treated press mud to the soil, by means of composting with other organic materials, is a more effective strategy for sustainable waste management.

The increased leaf area, together with the improved vegetative growth and dry matter production, can explain why, in the present research, the biological yield of sweet maize was significantly influenced by the two compost materials and by the sugarcane filter mud. The published literature suggested that a higher biological yield could occur through enhanced vegetative growth and dry matter accumulation when applying compost at various rates [50]. Other side factors could include enhanced disease protection, transplantation of photosynthetic material, turgor pressure management, and plant growth due to water conservation [51]. Moreover, our results are in accordance with Rana and collaborators [52], who reported that application of press mud led to the amelioration of the biological, physical, and chemical properties of soil, resulting in an improved supply

and uptake of nutrients, which ultimately led to better growth of plants and higher grain and biological yields. Furthermore, we found that the grain yield was significantly influenced by the compost source and sugarcane filter mud application; similarly to Shafi and colleagues [53], who affirmed that maize grain productivity increased when incorporating plant residues. Similarly, organic fertilization was found to provide enough N for high sweet corn yields [40], and yield increases with compost application have also been reported [54].

In sweet maize, yield components per plant include ears per plant, kernels per ear (rows per ear × kernels per row), and thousand-kernel weight. Due to the higher N availability, the number of grains per ear was significantly boosted by the composts and sugarcane filter mud. Similar results were reported by Ali and colleagues [55] using an organic nitrogen source. It was reported that combining organic and inorganic nitrogen increased the N uptake and thereby the number of grains per ear [55].

Likewise, in our research, the thousand-kernel weight was increased, in agreement with Younas and coworkers [56], and the CDA analysis further confirmed that thousand-kernel weight, grain yield, and biological yield were associated with sugarcane filter mud application. It was previously reported that applying press mud increased crop nutrient absorption, which in turn enhanced the growth, grain yield, and yield components of maize [57].

Ear weight and length were significantly affected, not only by the compost source, but also by SFM, consistently with Kumpawat and Rathore, as well as with Asif and colleagues [58,59], who revealed that the application of sugarcane filter mud resulted in a greater ear weight and increased ear length. These findings were in accordance with those of Gunjal et al. [60], who advocated press mud as a good organic manure, the application of which significantly increased the availability of nitrogen in the soil, resulting in improved yield attributes, such as ear length. It was also confirmed that the most effective dose of filter mud cake for improved ear weight was 2 t ha$^{-1}$ [61].

We also confirmed positive effects on the shelling percentage from the sugarcane filter mud and composts [42]; moreover, Iqbal et al. [54] reported that the maximum shelling percentage was observed with sugarcane filter mud application.

The harvest index was significantly boosted by the application of domestic waste and sugarcane filter mud. The CDA data confirmed that the harvest index was clearly linked to the use of composted household waste. Equally, Ajmal et al. [62] reported that press mud had a substantial impact on the harvest index, further confirming the findings of Muhammad et al. [38].

In sweet maize, yield components per plant include the ears per plant, kernels per ear (rows per ear, kernels per row), and thousand kernel weight, which are determined at different time points in the plant growth cycle. Thus, although we did not carry out soil analyses, our results implied that all the studied composts, from all sources, provided a consistent N availability during the entire crop cycle, even during ear development, grain filling, and in the late development phases (when the thousand-kernel weight was determined). Accordingly, we can conclude that sugarcane filter mud can be combined to compost to support sustainable sweet maize production; however, farmers should carefully select an appropriate local resource management for sweet maize, to sustainably produce the best kernel yields.

## 5. Conclusions

At present, the reutilization of the by-products generated from agro-industry represents a key strategy for improving the sustainability of agro-ecosystems, from economic and environmental points of view. However, the application of by-products should be site-specifically evaluated, to assess whether they can form a supplement or substitute chemical fertilizers in a sustainable agronomic management. Studies comparing local organic sources with sugarcane filter mud addition are lacking. In this study, the effects of different composted organic sources (agricultural or domestic wastes, i.e., brassica or

household residues, respectively) applied in combination with sugarcane filter mud on the growth and production of sweet maize were evaluated. We demonstrated that the domestic wastes performed better than the brassica residues, likely owing to better physical properties and to greater contents of readily available nutrients. In addition, the application of sugarcane filter mud at 2 t ha$^{-1}$ further improved the crop establishment with both composted materials, producing taller plants with larger leaves and higher grain and biological yields, as a result of more grains per ear and heavier grains. Thus, the results clearly showed that a combination of sugarcane filter mud and domestic compost was the most effective for integrating conventionally managed N fertilization, in terms of yield and other plant parameters. Thus, although further studies are needed to evaluate the long-term effects of sugarcane filter mud application on sweet maize and other crops, together with the physical–chemical effects and microbiological soil fertility, we have preliminary demonstrated that (i) combining sugarcane filter mud with composted domestic waste can be a successful fertilization approach for improving sweet maize yields in poor soils, and (ii) local industrial wastes can be effectively used for sweet maize cultivation, enhancing the sustainability of production. These results are pioneering for the development best management practices for the incorporation of sugarcane filter mud as an organic fertilizer and a means of waste disposal.

**Author Contributions:** Conceptualization, M.S., S.P. and A.J.; methodology, I. and M.S.; software, A.M., A.J. and M.F.S.; validation, A.J., M.F.S., S.P. and A.M.; formal analysis, E.R., S.P. and A.J.; investigation, M.S. and J.U.; resources, M.F.S., A.J. and I.; data curation, A.J., E.R., S.P., A.N. and M.F.S.; writing—original draft preparation, M.S., M.F.S., A.M., S.P. and A.J.; writing—review and editing, A.J., A.M., M.F.S., E.R., I.A. and S.P.; visualization, S.P.; supervision, I.; project administration, I.; funding acquisition, S.P., A.J. and M.F.S. All authors have read and agreed to the published version of the manuscript.

**Funding:** This research received no external funding.

**Institutional Review Board Statement:** Not applicable.

**Informed Consent Statement:** Not applicable.

**Data Availability Statement:** Not applicable.

**Acknowledgments:** We are thankful to the laboratory staff of Department of Agronomy, Faculty of Crop Production Sciences, The University of Agriculture, Peshawar (Pakistan) for their assistance and technical support during this work. We gratefully acknowledge Muhammad Fawad (Department of Weed Science and Botany, Faculty of Crop Protection Sciences, The University of Agriculture, Peshawar 25130, Pakistan) for his technical help throughout the writing of this manuscript. The authors thank the Higher Education Commission (HEC) of Pakistan for financial support. This study was also financially assisted by HEC, Pakistan, under HEC Project. Ref No. 20-15833/NRPU/R&D/HEC.

**Conflicts of Interest:** The authors declare no conflict of interest.

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
