# Peer review of "Composting Sugarcane Filter Mud with Different Sources Differently Benefits Sweet Maize"

_agronomy, doi:10.3390/agronomy13030748_

Round 1

Reviewer 1 Report

The research work is attractive due to the incursion of new nutritional sources for food production at a global level. Therefore, I recommend the authors continue with the line of research and the affectations or benefits that organic contributions bring to calcareous soils, making more references and studies to the ground with quantify the physical, chemical, biological, and micromorphological changes.

Reviewer 2 Report

The manuscript of Agronomy No. 2165233 with the title "Organic source matters: compost from domestic wastes benefits sweet maize more than agricultural residues, and further with sugarcane filter mud addiction", written by Muhammad Salman et al., should be accepted for publication after mayor revision.

General comments:

The article is related with the agronomic valorization of organic wastes such are compost produced from household and agricultural wastes and sugarcane filter mud, which is an interesting topic for the journal Agronomy as filter mud constitutes an important environmental problem. Its application in compost amendments in sweet corn production could be an interesting option in an international context as both, filter mud and corn are widely produced in many countries.

However, before its acceptance for publication different aspects must be changed or clarified:

-       -   In my opinion the experience is not correctly described as in line 150 it seams that composting process is carried out with and without SFM, but any ratio of the different components (SFM, household or crop wastes) is indicated. The information of the composting process is scarce without information about its duration, temperature reached and the quality of the final product.

-       -   SFM characterization needs to be included. Some properties, such as Total Carbon content, have very low content in CS2+SFM) but this is not reflected in OM contents.

-      -    In the results section appears the CS0 treatment without a proper description in the experimental design. Is it the control soil? Why is it introduced a treatment directly with SFM if the SMF in other treatments is introduced in the composting process?

-       -   The crop monitoring is carried out exhaustively and perfectly described, but some basic soil properties and its evolution are needed.

-        -  It is not clear the benefits of adding SFM to the compost as more of its effects on plant performance properties are not significant. This is not clearly expressed in the conclusions of the work.

-        -  The work must be put in a more international context. The bibliographic section needs to be revised as many of the references, mainly in the discussion section are only given in the context of Pakistan.

Reviewer 3 Report

The manuscript presents a one-year field study - in my opinion, this is not enough to publish such results in a journal like Agronomy. Field trials should last at least three years. I think sometimes a two-year study is acceptable, never a one-year study. In addition, the title of the  manuscript is very bad, unacceptable. The methodology of the study is misrepresented, because what does it mean: "The study was conducted in a factorial design in triplicate with the following treatments: [household waste (CS2)] applied at a rate of 9 t ha-1, ii) two applications of sugarcane filter slurry (SFM) [SFM applied at a rate of 2 t ha-1 (+SFM), and SFM not applied (-SFM)]. In experiments of this type there are always factors, it is necessary to specify whether one or two.
Experimental objects were described (text above) and in my opinion it was done wrong. There were no "two Compost Sources" only composts made from different materials with or without the addition of "Sugarcane Filter Mud" . This is clear from Tab. 1. This is what I understood from the text presented in the manuscript, which states that four different composts were tested in a field experiment. Therefore, the description of the results should be carried out in accordance with what I wrote, and this is not the case.
The soil conditions in which the research was carried out were not described in detail: "extractable phosphorous" - what method was used to extract P and the content of exchangeable K - in my opinion, it was assimilable potassium.
It should be noted that the samples of plant material were not very representative: five corn cobs or five plants. Meteorological conditions (precipitation and temperatures) were also given, and it was written that sprinkler irrigation was carried out. So what if no irrigation data was provided. I also have reservations about the data presented in Table 1. How was the content of "total carbon (%)" greater than the content of "total organic matter (%)" in CS1-SFM and CS1+SFM?

Round 2

Reviewer 2 Report

The authors have improved adequately the text, according to the suggestions of reviewers.

Minor changes_

- line 36: added text repeated "two controls"

- the classification of the soil would be Calcaric Fluvisol (FL ca)

- cited references in the text must include authors' names. E.g. in line 424, "was also reported by Kumar and Chopra [24]".

In general, the work is good and discussed exhaustively the effects of compost and SFM on crop performance, but the evolution of soil properties would improve the paper.

Reviewer 3 Report

I'm still thinking about the title of the manuscript. Perhaps a title along the lines of: 

Possibilities of using sugarcane filter mud to produce composts useful for fertilizing sweet maize

or Effect of compost with sugarcane filter mud on sweet maize

or Effect of different composts with sugarcane filter mud on sweet maize
